# KNOWLEDGE LIFT ALIGNMENT FINE TUNING

## ABSTRACT

We present a visual tuning framework, **K**nowledge **L**ift **A**lignment **F**ine **T**uning (KLAFT), which enhances the expressive image captioning capabilities of Pre-trained Language Models (PLMs), including LLMs and VLMs. As this task involves generating more detailed and comprehensive captions than basic image descriptions, the core idea behind KLAFT is that fine-grained alignment could exploit the capabilities of PLMs and a given target domain dataset. This idea motivates and challenges us to explore the framework that deeply understands both given images and text for this alignment and tuning PLMs towards expressive image captioning. This motivation modifies the attention mechanism to a Modified Attention Mechanism (MAM) and develops both a Topic Control Mechanism (TCM) and their training objectives. The innovation of KLAFT lies in its approach to addressing the disparities in knowledge - visual versus textual via MAM and source versus target domain via TCM. As these hidden spaces are conceptualized as distinct sub-networks within the PLM, each possessing specific knowledge, KLAFT's unique contribution is in aligning and adjusting the weights of these sub-networks in a fine-grained manner, and fine-tuning this PLM. Our empirical studies demonstrate that KLAFT significantly improves expressive captioning tasks by aligning and amplifying target knowledge, with the potential for Parameter-Efficient fine tuning (PEFT) at low computational cost.

## 1 INTRODUCTION

Our goal is to create descriptive and informative captions that address the limitations of models relying solely on visual features. In the field of computer vision, transformer-based models Vaswani et al. (2017); Devlin et al. (2019); Radford et al. (2019); Yang et al. (2019); Liu et al. (2019); Lan et al. (2020); Touvron et al. (2023); AI@Meta (2024) have demonstrated significant potential as PLMs, including Vision-Language Models (VLMs) Liu et al. (2023a), as demonstrated by various studies. However, these models often overlook additional knowledge individuals use when generating captions, beyond the inherent content of the data Chen et al. (2019). Source-domain datasets are sufficient for acquiring linguistic and source-domain knowledge but inadequate for learning target-domain knowledge. This robust prior knowledge may overshadow target domain-specific knowledge, rendering it ineffective. Interestingly, the robust prior knowledge that enhances the linguistic abilities of PLMs may also overshadow the target domain-specific knowledge, rendering it ineffective Ramasesh et al. (2021).

To guide PLMs in generating expressive descriptions, we introduce an adaptation framework called **K**nowledge **L**ift **A**lignment **T**uning (KLAFT). Unlike existing Knowledge Distillation methods Hinton et al. (2015), knowledge injection approaches Xu et al. (2023); Zhang et al. (2023), LoRA Hu et al. (2022), and adapters Hu et al. (2023); Zhang et al. (2024), KLAFT perceives hidden spaces as sub-networks within PLMs and aligns the weights of these sub-networks from the source to the target domain through fine-grained knowledge alignment and lift. This approach modifies the attention mechanism to a Modified Attention Mechanism (MAM) and develops a Topic Control Mechanism (TCM) with specific training objectives. KLAFT uses the concept of topics to bridge the "knowledge differences" in visual versus textual data via MAM, and source versus target domain by exploiting the hidden spaces within PLMs via TCM. The novelty of KLAFT lies in using knowledge to quantify the domain gap and introduce it into fine-tuning. To leverage knowledge, KLAFT aligns and adjusts the weights of these sub-networks in a fine-grained manner, fine-tuning the PLM towards expressive image captioning. This framework not only enhances knowledge adoption and tuning

Table 1: Example of image captions: The green highlighted words are detected via knowledge lift after knowledge alignment in KLAFT.

| Image | Model or Framework: generated texts |
|---|---|
|  | BLIP: Dog standing in a car stopping on the street
KLAFT: A golden retriever is waiting for its master in a truck. |

but also coexists with other PEFTs and fine-tunes various PLMs Radford et al. (2021); Jia et al. (2021); Li et al. (2023b); Liu et al. (2023a).

As with a visual tuning framework, our experiments confirm that KLAFT and its components:
1. Capture fine-grained interactions between images and texts through knowledge alignment and reflect the target domain knowledge via knowledge lift, as confirmed by human evaluations.
2. Demonstrate model-agnosticism, cooperating with other PLMs and VLMs, exploiting their capabilities, with low computational cost.

## 2 PREVIOUS WORK

There is increasing interest in vision-language tasks, such as image captioning Xu et al. (2015); Herdade et al. (2019); He et al. (2020); Cornia et al. (2020); Song et al. (2021); Liu et al. (2021); Chen et al. (2021a; 2022); Zhou et al. (2021). VisualGPT Chen et al. (2022) leverages the linguistic knowledge of the PLM, and introduces an encoder-decoder attention mechanism to bridge the differences between modalities. However, the structure between the textual units in images (usually the regions detected by the object detection model) and sentences (each single word) are different He et al. (2020). Wang et al. Wang et al. (2020) explicitly engage human consensus to measure the quality of ground truth captions in advance to resolve grammatical errors, wrong identification of visual objects, and sub-optimal sentence focus. For controllable image captioning, Chen et al Chen et al. (2021b) propose Verb-specific Semantic Roles, each of which consists of a verb and several semantic roles. CGRL Zhang et al. (2021) attempts to reproduce the human inference procedure, i.e., consensus graph representation learning framework, in the grounded captioning pipeline and model training. CAAG Song et al. (2021) guides the captioning model to learn semantics by reproducing the current generation based on the global contexts; it takes advantage of global predictions in this process. Xu et al. (2021) proposed an Anchor-Captioner to generate multiple captions from different views and so extract more valuable scene information. Ji et al. (2021) introduced a Global Enhanced Transformer to enable the extraction of more comprehensive global representations, and guide the decoder to generate high-quality captions. DLCT Luo et al. (2021) aims to realize the complementary advantages of region and grid features for image captioning.

Recently proposed prefix style models Mokady et al. (2021); Wang et al. (2022b); Tsimpoukelli et al. (2021); Zhou et al. (2022); Muresan et al. (2022) address this task by prepending image feature sequences to the text sequences. CLIP (Contrastive Language-Image Pre-Training) Radford et al. (2021)is a neural network that projects both images and texts into the same space and learns their representations. ClipCap Mokady et al. (2021) uses CLIP as the vision encoder, and maps the CLIP embedding as a prefix to the caption. ALIGN Jia et al. (2021) presents an example of leveraging large-scale noisy image-text data to strengthen visual and vision-language representation learning, while training a dual-encoder model with contrastive loss. COTS Lu et al. (2022a) addresses the high computation cost of object detection in images by proposing two-stream vision-language pre-training that leverages three levels of cross-modal interactions in image-text retrieval, while still maintaining its advantage of efficiency for image-text retrieval. BLIP Li et al. (2022) pre-trains a multi-modal mixture of encoder-decoder models using a dataset bootstrapped from large-scale noisy image-text pairs by injecting diverse synthetic captions and removing noisy captions. CoCa Yu et al. (2022) also uses CLIP as the vision encoder, and pretrains an image-text encoder-decoder foundation model jointly with contrastive loss and captioning loss.

Given that the catastrophic forgetting problem implies a loss of PLMs' knowledge by overwriting their parameters through fine-tuning Mccloskey & Cohen (1989); Mi et al. (2020), KLAFT tackles this problem by collaborating their source knowledge with the target knowledge, and belongs to PEFT Hu et al. (2022); Wang et al. (2022a); Li et al. (2023c); Anonymus (-a;b). Unlike knowledge injection, "knowledge lift" aims to highlight the target domain inherent in PLMs, and allows KLAFT

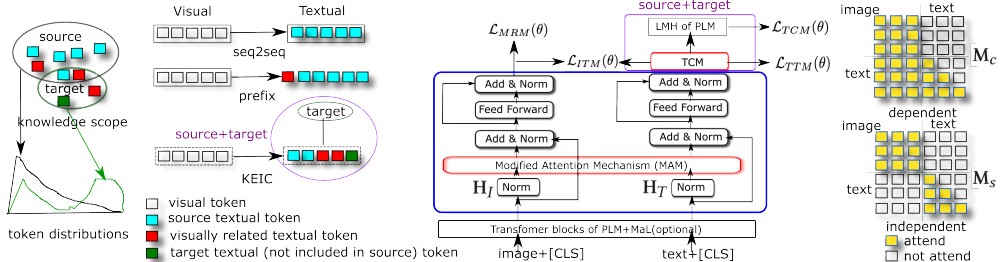

Figure 1: (left) Differences between source and target, (center left) The motivation of KLAFT is to consider differences between visual and textual, and reflect more target tokens in captions, and textual, than the others, as shown in Table 1, (center right) Architectures of KLAFT: KLAFT plugs a Modified Attention Mechanism (MAM) (blue box), TCM, and training tasks MRM, ITM, and TTM, into the base PLM (Transformer blocks + Language model head (LMH)), where MAM aligns visual or textual on the token level, and TCM highlights the target knowledge through the knowledge lift. (right) Masks of MAM: During the training phase, KLAFT uses the dependent mask, $M_c$, to maximize the probability of both the masked features and the next token. On the inference, KLAFT uses an independent mask, $M_s$, to yield the text for a given image.

to adopt the visual-textual space, VLMs (e.g., BLIP2 Li et al. (2023b) and Llava Liu et al. (2023a)), or collaborate with other PEFT (e.g., LoRA and MixPHM), knowledge injection, prompt tuning, or in-context learning.

## 3 KNOWLEDGE LIFT ALIGNMENT FINE TUNING (KLAFT)

### 3.1 BASIC IDEAS, DEFINITIONS, AND ARCHITECTURE

In image captioning, the absence of robust metrics for the automatic and quantitative evaluation of fine-grained granularity in captions is a notable gap. We posit that qualitative evaluations serve as accurate reflections of the knowledge inherent in the target domain, and that fine-grained models, as exemplified in Table 1, could generate detailed captions. As shown in Figure 1(left), the source data is used to train PLMs, while the target data is utilized to fine-tune these PLMs. To manage the disparity between both domains, we implement domain adaptation tuning, which can provide further insights such as the distribution of tokens used.

KLAFT, seq2seq Cornia et al. (2020); Huang et al. (2019); Chen et al. (2022) and prefix-based image captions Mokady et al. (2021); Tsimpoukelli et al. (2021); Wang et al. (2022b), as shown in Figure 1(center left), comprise a vision encoder, which converts an image into visual information, and a text decoder. KLAFT focuses on the sub-networks within PLMs, each of which encapsulates specific knowledge influencing token distribution and token embedding representation.

KLAFT's concept is the "knowledge differences", which can be source vs target domain and visual vs textual, can be bridged by fine-grained alignment, thereby enhancing the explanatory power of the generated captions. This idea motivates the design of a Mapping Layer (MaL) and a Modified Attention Mechanism (MAM) for **knowledge alignment**, as illustrated in Figure 1(center right). We interpret knowledge as hidden representation $\mathbf{h}_{*,*}$, and domain-specific knowledge as domain-specific tokens. We employ latent discrete variables to detect hidden representations as sub-networks and define their weight in the PLM. This approach maintains a global statistical view akin to topic models and adapts the PLM by adjusting this weight to match the target data.

To generate expressive descriptions, we design a Topic Control Mechanism (TCM) to emphasize the target domain-specific knowledge. Since the motivation for this framework is its application to diverse PLMs, we place both MAM and TCM between the top layer of the Transformer blocks and the LM Head. TCM distinguishes this knowledge from the rest, making it more accessible for exploitation in training tasks such as Token Topic Modeling (TTM), Masked Region Modeling (MRM), and Text Image Matching (TIM). This mechanism allows KLAFT to highlight the target-related knowledge, the **knowledge lift**. By employing these tasks and different masks shown in Figure 1(right), KLAFT guides PLMs to generate texts that encompass all elements (i.e., visual, source, and target), thereby reflecting the target knowledge in expressive text.

### 3.2 VISION ENCODER AND MAPPING LAYER (MAL)

Mapping Layer (MaL) projects the output of the vision encoder into the same space as the output of the text decoder so that the text decoder can understand this output like the textual tokens of this decoder. Recent VLMs Li et al. (2023b); Liu et al. (2023a) already do this, so MaL is an option. To apply various image object detection networks or other VLMs, the vision encoder is designed to accept both methods and work with the text decoder seamlessly. We call the length of the final output of the vision encoder, the hidden representations, $\mathbf{H}_L \in \mathbb{R}^{I_{pf} \times d_{pf}}$, where $\mathbf{H}_L$ is visual **knowledge**, $I_{pf}$ is the prefix length and $d_{pf}$ is the dimension size.

MaL is placed on the vision encoder to map hidden representation vector $\mathbf{H}_L = [h_{L,1}, \cdots, h_{L,I_{pf}}] \in \mathbb{R}^{I_{pf} \times d_{pf}}$ to give it the same dimension as the size of token embedding, $d_h$. Motivated by the well-known mapping methods of deep-learning schemes, we propose three candidates for mapping functions of MaL and embed the visual tokens $\mathbf{H}_I \in \mathbb{R}^{I_{pf} \times d_h}$ as follows:

$$\mathbf{H}_I = \begin{cases} \mathbf{h}_{L,t}\mathbf{W}_{pf} & \text{simple transform for } \mathbf{h}_{L,t} \in \mathbf{H}_L \\ MLP_l(\mathbf{H}_L) & \text{MLP if } I_{pf} > 1 \\ Tra_l(\mathbf{H}_L) & \text{Transformer if } I_{pf} > 1, \end{cases} \tag{1}$$

where $\mathbf{W}_{pf} \in \mathbb{R}^{d_{pf} \times d_h}$ are learnable weights, $MLP_l()$ is the multi-layer perceptron with $l$ layers, and $Tra_l()$ is the bi-directional Transformer with $l$ layers. When $I_{pf} > 1$, our pre-ablation analysis confirms that $MLP_l()$ attains better performance at a lower cost than $Tra_l()$. KLAFT optimizes the image embeddings by aligning visual and textual in the same space through ITM.

### 3.3 MODIFIED ATTENTION MECHANISM (MAM)

To clarify the differences between visual and textual knowledge on the textual level of given image-text pairs, we modify a self-attention mechanism, MAM, to control the attention between visual features and text as illustrated in Figure 1(right). The text decoder is responsible for generating the next token of the output caption, conditioned on both previously generated tokes and visual tokens, $\mathbf{H}_I$, converting the text sequence into the text data, and feeding it, $\mathbf{H}_0 = [e_1, \cdots, e_{|x|}]$, to the next layer. It yields the final output, $\mathbf{H}_l \in \mathbb{R}^{d_h \times |x|}$, textual **knowledge**, like the vision encoder, where $d_h$, and $|x|$ are the hidden dimension, and the number of textual tokens, respectively. While visual tokens and textual tokens can influence other tokens like cross-attention, they should exercise influence only with tokens of the same type like self-attention. In order to balance this influence, MAM unifies two different attention masks, $\mathbf{M}_c, \mathbf{M}_s \in \mathbb{R}^{(I_{pf}+|x|) \times (I_{pf}+|x|)}$, as illustrated in Figure 1(right).

This balance is automatically controlled by the gate matrix, $\mathbf{B}_c \in \mathbb{R}^{(I_{pf}+|x|) \times (I_{pf}+|x|)}$, to set the relative strengths of the vision encoder layer, $\mathbf{H}_I$, and text decoder layer, $\mathbf{H}_T$ (=$\mathbf{H}_L$), on top of each layer. We gain the query, key, value, $\mathbf{Q}, \mathbf{K}, \mathbf{V}$, from $\mathbf{H}_I \oplus \mathbf{H}_T$ like other Transformer models, and define MAM using masks shown in Figure 1(right) as:

$$MAM(\mathbf{Q}, \mathbf{K}, \mathbf{V}) = \mathbf{B}_c \otimes softmax(\frac{\mathbf{Q}\mathbf{K}^T}{\sqrt{d_k}} + \mathbf{M}_c)\mathbf{V} + (\mathbf{1} - \mathbf{B}_c) \otimes softmax(\frac{\mathbf{Q}\mathbf{K}^T}{\sqrt{d_k}} + \mathbf{M}_s)\mathbf{V},$$

$$\mathbf{B}_c = \sigma(\mathbf{A})\mathbf{I}(\sigma(\mathbf{A}) > \mu), \mathbf{A} = [\mathbf{H}_I \oplus \mathbf{H}_T]\mathbf{W}_b + \mathbf{C}_b, \tag{2}$$

where $\otimes$ denotes component-wise multiplication, and $\mathbf{W}_b \in \mathbb{R}^{d_h \times (I_{pf}+|x|)}$ and $\mathbf{C}_b \in \mathbb{R}^{(I_{pf}+|x|) \times (I_{pf}+|x|)}$ are learnable weights, $\mu$ is gate threshold value, and $\mathbf{I}$ is the indicator function that returns 1 if the inner statement is true and 0 otherwise. This function aims to mitigate overfitting by suppressing small values and inducing sparse activation. It applies bidirectional attention between visual and textual tokens, where, $M_*(i,j) \in \mathbf{M}_* = 0$ allows the $i$-th position to attend to the $j$-th position whereas $M_{ij} = -\infty$ prevents it from attending. KLAFT represents both visual and textual information in the same space through MAL, and feeds $MAM(\mathbf{Q}, \mathbf{K}, \mathbf{V})$ to a feedforward layer with ReLU activation Nair & Hinton (2010).

While the Modified Attention Mechanism shares conceptual similarities with those commonly seen in VisualGPT Chen et al. (2022), vision-language models (VLMs), it not only aligns images and text but also differentiates between visual versus textual and source versus target domains. This dual alignment enables more precise internal knowledge extraction, crucial for enhancing model performance. The mathematical formulation, although inspired by traditional self-attention, is uniquely tailored to address these dual alignment challenges, thereby offering a novel contribution to the field.

### 3.4 Topic Control Mechanism (TCM)

TCM is designed to highlight the target domain instead of overwriting the parameters of PLMs. Unlike PEFTs, TCM is placed just below the token verbalizer to model the uncertainty in the generative process, after mitigating the discrepancy between the source and the target domain knowledge. This placement ensures that TCM captures the contextual interdependencies between tokens.

In NLP, language models are trained for tasks that require text generation Bengio et al. (2003). Given text sequence $\mathbf{x}_d = \{x_{d,1}, \cdots, x_{d,|x_d|}\}$ and dataset $D = \{\mathbf{x}_1, \cdots, \mathbf{x}_D\}$, models are pre-trained by maximizing the likelihood under forward autoregressive factorization. Since we focus on the differences between the source and target, and the difference in their distributions, as shown in Figure 1(left) and explained in 3.1, KLAFT trains PLMs by moving these distributions closer to the distributions observed in the target domain, i.e., topic lift. This text generation process is given by:

$$\mathcal{L}_{TCM}(\theta) = -\sum_{d=1}^{|D|} \sum_{t=1}^{|x|} \log \sum_{z_t=1}^{K} P_\theta(x_{d,t}|\mathbf{x}_{d,1:t-1}, z_t) P_\theta(z_t|\mathbf{x}_{d,1:t-1}), \tag{3}$$

where $\theta$ represents model parameters and $z_t$ indicates the topic of the $t$-th token, $\mathbf{K}$ is the number of topics, $P_\theta(z_t|\mathbf{x}_{d,1:t-1})$ is the prior distribution over topic $z$, and $P_\theta(x_{d,t}|\mathbf{x}_{d,1:t-1}, z_t)$ is the "generative" distribution over tokes, $\mathbf{V}$. In Figure 1(left), the ratio of domain (top), and the distribution (bottom) corresponds to $P_\theta(z_t|\mathbf{x}_{d,1:t-1})$, and $P_\theta(x_{d,t}|\mathbf{x}_{d,1:t-1}, z_t)$, respectively. This formulation ensures that text $\mathbf{x}_d$ can be generated by a random process involving $z$: (1) $z_t$ is first generated from the conditional distribution $P_\theta(z_t|\mathbf{x}_{d,1:t-1})$. (2) $x_{d,t}$ is finally generated from $P_\theta(x_{d,t}|\mathbf{x}_{d,1:t-1}, z_t)$. Since global distributions do not require additional learning, TCM finds target-specific distributions through topics, and updates them, $P_\theta(z_t|\mathbf{x}_{d,1:t-1})$ and $P_\theta(x_{d,t}|z_t, \mathbf{x}_{d,1:t-1})$, in fine-tuning.

Following Eq (3), TCM maps hidden representation vector $\mathbf{H}_L = [h_{L,1}, \cdots, h_{L,|x|}] \in \mathbb{R}^{d_h \times |x|}$, **knowledge**, onto topic vector $\mathbf{z} \in \mathbb{R}^K$, and then projects this topic vector into the topic-specific distribution over tokes. This yields Eq (3) by defining topic matrix, $\mathbf{W}_{TZ} \in \mathbb{R}^{d_h \times K}$, and word generation function, $\mathcal{F}(\mathbf{h}_{L,t}, z_t)$, where $V$ is vocabulary size. We apply these matrices to $\mathbf{h}_L, t \in \mathbb{R}^{|x| \times d_h}$ in the text decoder. Given the above, we gain $P_\theta(z_t|\mathbf{x}_{d,1:t-1})$ and $P_\theta(x_{d,t}|\mathbf{x}_{d,1:t-1}, z_t)$ as follows:

$$P_\theta(z_t|\mathbf{x}_{d,1:t-1}) \propto Softmax(LayerNorm(\mathbf{h}_{L,t})\mathbf{W}_{TZ}), \quad P_\theta(x_{d,t}|\mathbf{x}_{d,1:t-1}, z_t) \propto \mathcal{F}(\mathbf{h}_{L,t}, z_t) \quad (4)$$

where $\mathbf{W}_{TZ}$ are learnable weights. As with $\mathcal{F}(\mathbf{h}_{L,t}, z_t)$, we use the existing LMH of PLMs. Motivated by the conventional activation functions of deep learning, three transformations (e.g., addition, multiplication, and affine), we branch off $\mathbf{h}_{L,t}$ and convert it as the only network in the subnetworks:

$$\mathcal{F}(\mathbf{h}_{L,t}, z_t = z) = \mathbf{LMH}(\mathbf{h}_{L,t}), \quad \mathbf{h}_{L,t} = \begin{cases} \mathbf{h}_{L,t} & \text{residual if } z = 0 \\ (1-\omega)\mathbf{h}_{L,t} + \omega\mathbf{g}_z & \text{addition if } z > 0 \end{cases} \tag{5}$$

where $\mathbf{LMH}()$ is the language model head of the base PLM, $\mathbf{g}_z \in \mathbb{R}^{d_h}$, $\mathbf{W}_z \in \mathbb{R}^{d_h \times d_h}$, and $\mathbf{b}_z \in \mathbb{R}^{d_h}$ are the topic $z$ specific learnable weights. To preserve the source knowledge of PLM, TCM is designed to retain the pre-trained hidden representations, denoted as $\mathbf{h}_{L,t}$, when $z = 0$, instead of overwriting the parameters, otherwise facilitates the accumulation of gradient updates throughout the fine-tuning process, similar to the approach used in LoRA Hu et al. (2022). However, it differs in 1) using only $\mathbf{h}_{L,t} \in \mathbf{H}_L$ in Eq (5), 2) shifting the weight rather than dimensionality reduction. Our ablation analysis demonstrated that the addition operation yielded the best performance over functions ($\mathbf{h}_{L,t} \otimes g_z$ multiplication, $\mathbf{h}_{L,t}\mathbf{W}_z + \mathbf{b}_z$ affine), leading us to incorporate it into Eq (5). The transformation of Eq (3) is facilitated by the introduction of the topic matrix. The previous Transformer-block can be decomposed into the product of $\mathbf{W}_{TZ}$ and $\mathcal{F}(\mathbf{h}_{L,t})$ as shown in Eq (4). While these subnetworks are akin to the concept of Mixture of Experts (MoE) Szymanski & Lemmon (1993), where different combinations of model blocks are utilized depending on the task, KLAFT adds this functions, $(1-\omega)\mathbf{h}_{L,t} + \omega\mathbf{g}_z$ ($z > 0$), to an existing network, $\mathbf{h}_{L,t}$ ($z = 0$), rather than creating new networks, where the number of topics corresponds to the number of experts.

While topics can be considered as a quantized sample of the underlying token distribution, we define Eq (5) as a differentiable function that is end-to-end learnable together with training (i.e., backpropagation with cross-validation over tokens and training tasks). This ensures that KLAFT can update topic-related parameters through other hidden parameters. Collapsed Gibbs sampling, which is

employed in topic models, needs higher computational cost and trick used in the Variational AutoEncoder (VAE) Kingma & Welling (2014). As the blue token could be replaced by the green token in Figure 1(center left), this design allows PLMs to highlight knowledge that might otherwise have been suppressed, thus preventing catastrophic forgetting Mccloskey & Cohen (1989); Mi et al. (2020).

# 4 TRAINING OBJECTIVES OF KEIC

**Token Topic Modeling (TTM)**: Motivated by the masked language model (MLM) of Vaswani et al. (2017), this training task allows topic $z$ to capture the contextual interdependence between tokens in the decoder, while avoiding obvious local optima when the decoder simply generates latent vectors that encode only the corresponding token. To avoid this undesired local optimum, we randomly apply token-level dropout to an entire token when computing the posterior. This technique ensures that the model has to learn how to use contextual information. Different from MLM, TTM employs $\mathbb{L}^2$ instead of the log-likelihood to measure the similarity between tokens on topics.

$$\mathcal{L}_{TTM}(\theta) = \sum_{d=1}^{|D|} \sum_{t \sim x_d} ||\mathbf{h}_{L,t} - \tilde{\mathbf{h}}_{\mathbf{L,t}}||_2^2, \quad \tilde{h}_{L,t} = (1-\omega)\mathbf{h}_{L,t} + \omega\mathbf{g}_z, \tag{6}$$

where $\mathbf{h}_{L,t}$, $\mathbf{g}_z$, and $\omega$ are shown in Eq (5). When $\omega = 0$, the pre-ablation analysis showed that we could minimize only $\mathcal{L}_{TTM}$, but not optimize $\mathcal{L}_{KLAFT}(\theta)$ in Eq (9). Then, this value results in reducing the overall performance, as shown in Table 4 and Table 5.

**Masked Region Modeling (MRM)**: As each image is converted into a sequence of visual regions, $\mathbf{v_k} = \mathbf{v_{k,1}}, \cdots, \mathbf{v_{k,I_{pf}}}$, like textual tokens, we apply MRM Chen et al. (2020) if $|I_{pf}| > 1$. We denote the image region, and the mask indices as $m \in \mathbb{N}^M$, and randomly sample visual regions and mask out these regions with the probability of 15%. KLAFT is trained to predict the masked regions based on observations of their surrounding regions $\mathbf{v_{k,\backslash m}}$, by minimizing this function:

$$\mathcal{L}_{MRM}(\theta) = E_{(\mathbf{v}) \sim \mathcal{I}} f_\theta(\mathbf{v_{k,m}}|\mathbf{v_{k,\backslash m}}), \quad \mathbf{f}_\theta(\mathbf{v_{k,m}}|\mathbf{v_{k,\backslash m}}) = ||\hat{\mathbf{v}}_{\mathbf{k,m}} - \mathbf{v_{k,m}}||_\mathbf{2}^\mathbf{2}, \tag{7}$$

where $\hat{\mathbf{v}}_{k,m}$ corresponds $h_{I,m}$ in Eq (1), and $f_\theta$ is the multi-layer perceptron. The error between the actual visual and the predicted feature is quantified to optimize the parameters of MaL.

**Image Text Matching (ITM)**: The objective of ITM is to learn the relationship between images and texts at the instance level. Unlike other ITM of UNITER Chen et al. (2020) and ViLBERT Lu et al. (2019), we employ a triplet objective-based function to evaluate the similarities between images and texts, and use [CLS] as shown in Figure 1. The special token, [CLS], which is used by Transformer-encoder based models to encode a given input as a whole and gain its representation. However, the text decoder does not have this special token. To overcome this issue, KLAFT appends [CLS] to the end of each image and text, respectively. This addition ensures that KLAFT can obtain an instance-level representation of the image, $\mathbf{v}_a$, and its text, $\mathbf{v}_x$, respectively, and can apply the contrastive loss.

Given image vector $\mathbf{v}_a$ as an anchor attribute embedding, and its corresponding text vector, $\mathbf{v}_x$, as a positive embedding, and the other text vector, $\mathbf{v}_y$ as a negative embedding, triplet loss tunes the model such that the distance between $\mathbf{v}_a$ and $\mathbf{v}_x$ is smaller than the distance between $\mathbf{z}_a$ and $\mathbf{v}_y$. Mathematically, this objective minimizes the following loss function:

$$\mathcal{L}_{ITM}(\theta) = \max_{(\mathbf{v}_a, \mathbf{v}_x, \mathbf{v}_y) \sim \mathbf{B}} (||\mathbf{v}_a - \mathbf{v}_x|| - ||\mathbf{v}_a - \mathbf{v}_y|| + \epsilon, 0), \tag{8}$$

where $\mathbf{B}$ is each batch, $|| \bullet ||$ is a distance metric and $\epsilon$ is the margin that ensures that $\mathbf{v}_x$ is at least $\epsilon$ closer to $\mathbf{v}_a$ than $\mathbf{v}_y$, and the sampling targets are batch units. We compared two loss functions, the triplet objective-based function and the contrastive loss function, in preliminary experiments, and confirms that the former outperformed the latter. The MaL focuses on transforming input features to align with the model's requirements, enhancing learning and generalization. In contrast, contrastive learning is a self-supervised technique that learns representations by contrasting positive and negative pairs. This is an important task as it helps to bridge the differences between visual and textual knowledge on the instance level, and optimize the parameters using two methods, MaL and TCM.

To optimize these parameters and close the difference between the pre-training and the fine-tuning process, KLAFT optimize the model loss in the tuning process. Using (6,7,8), we can insert Eq (3)

into the loss function, $\mathcal{L}(\theta)$, that is defined as the sum of these objective functions and is optimized in the fine-tuning stage:

$$\mathcal{L}_{KLAFT}(\theta) = \underbrace{\lambda_{MRM}\mathcal{L}_{MRM}(\theta)}_{encoder} + \underbrace{\lambda_{ITM}\mathcal{L}_{ITM}(\theta)}_{encoder-decoder} + \underbrace{\mathcal{L}_{TCM}(\theta) + \lambda_{TTM}\mathcal{L}_{TTM}(\theta)}_{decoder}, \tag{9}$$

where $\theta$ is the parameter set of KLAFT, $\lambda_{MRM}$, $\lambda_{ITM}$ and $\lambda_{TTM}$ are hyper parameters to balance the importance of MRM, ITM, and TTM, respectively. We adopt Adaptive Moment Estimation (Adam) (Kingma & Ba, 2015) over mini-batches, and the dropout strategy (Srivastava et al., 2014) to update parameter and optimize networks.

## 5 EXPERIMENTS

### 5.1 IMPLEMENTATIONS

We implemented KLAFT by using Pytorch 2.0.1[1] and will release this code soon. Through experiments, $\lambda_{MRM}$, $\lambda_{ITM}$ and $\lambda_{TTM}$ are set to 0,1, 0,1, and 0.1.

### 5.2 IMAGE CAPTIONING TASK

We compare it with baselines on the two datasets of MS COCO Lin et al. (2014), and Conceptual Captions Sharma et al. (2018) for PLMs. As the test data of COCO is not publicly available, we followed the settings used in Cornia et al. (2020); Huang et al. (2019); Chen et al. (2022), and converted all sentences to lower case. The validation data set was taken as our test set, and 5000 different image-caption pairs were randomly sampled from the training set as the validation set. To create the small training data setup for COCO and Conceptual Captions, we randomly sampled 0.1%, 1.0%, and 100.0% image-caption pairs as training data. The procedure was repeated 4 times with different random seeds.

We compare KLAFT with state-of-the-art models in 6 various settings, S1-6; we followed its setting (i.e., VisualGPT), and applied it (COCO+Conceptual Captions) to comparisons including baselines (e.g., AOA, M2 Transformer, and VisualGPT) in S1-S3. For S4-S6, we use only COCO.

**Automated evaluation**: To evaluate and compare models, we follow prior studies Saha et al. (2018); Cui et al. (2019), and use BLEU-N Papineni et al. (2002), METEOR Lavie & Agarwal (2007), ROUGE Lin (2004), CIDEr Vedantam et al. (2015), and SPICE Anderson et al. (2016) metrics to measure the performance. To evaluate the effect of TCM, we prepared the following metric;

$$r_{TCM} = \frac{1}{|\mathcal{X}_t|} \sum_{i \in \mathcal{X}_t} \frac{1}{|\mathbf{x}_i|} \sum_{j \in \mathbf{x}_i} z_{ij}, z_{ij} = \begin{cases} 0 & \text{if } z_{ij} = 0 \\ 1 & \text{else,} \end{cases} \tag{10}$$

where $\mathcal{X}_t$ is the set of test captions, $\mathbf{x}_i$ is the set of tokens in the $i$-th text, and $z_{ij}$ is the topic indicator in Eq (3). The larger this value is, more topics other than "residual" are selected for each token, as shown in Eq (3). Table 2 shows that the components of KLAFT enhance PLMs (e.g., GPT-2) in (S3 vs S1/S2, and S4 vs S3), and the increase in $r_{TCM}$ confirms that TCM successfully extracted the target-specific knowledge, and achieved better performance in both S3 and S4. S1-S3 shows that KLAFT is highly efficient in learning under scaled-down data, KLAFT only updates the representation associated with each topic, which allows it to coexist with PLM without contradicting the motivation behind it. KLAFT cooperates with the VLMs and contributes to their performance improvement, as shown in S5 and S6. The results of S4-6 indicate that TCM and TTM could be effective PEFT methods not only in improving computational efficiency but also in closing the knowledge differences.

**Human evaluation**: To compare the fluency of fine-grained captions, we conducted a fluency test with human annotation on text generated from 100 images randomly sampled from the test set. The results of KLAFT and the most competitive baseline models were compared. We recruited a diverse group of annotators, each asked to judge using fluency and adequacy as criteria. Fluency assesses whether a sentence is grammatically fluent, while adequacy measures the expressiveness of the caption.

---

[1]https://pytorch.org/

Table 2: Performance Comparison of image captioning models on image-caption pairs: In KLAFT, $\omega$ is defined in Eq (5), $K$ is #topics, AD used in $\mathcal{F}$. In the top row, B-1, B-4, M, R, C, and S denotes BLEU-1, BLEU-4, METEOR, ROUGE, CIDEr, and SPICE, respectively. The bold value denotes the statistically superior value, $p < 0.01$, compared to the best baseline.

| Model | B-1 | B-4 | M | R | C | S | $r_{TCM}$ |
|---|---|---|---|---|---|---|---|
| S1: vs seq2seq under fine-tuning GPT-2 over 0.1% training data (COCO+Conceptual Captions) | | | | | | | |
| Transformer Vaswani et al. (2017) | 54.4 | 14.8 | 16.5 | 36.0 | 44.4 | 7.8 | - |
| $\mathcal{M}^2$ Transformer Cornia et al. (2020) | 55.0 | 14.8 | 16.7 | 39.5 | 43.1 | 7.8 | - |
| AOA Huang et al. (2019) Transformer | 56.7 | 14.5 | 16.9 | 34.0 | 41.0 | 7.2 | - |
| DLCT Luo et al. (2021) | 55.5 | 14.7 | 16.9 | 39.3 | 43.6 | 7.7 | - |
| SATIC Zhou et al. (2021) | 55.1 | 14.5 | 15.6 | 39.1 | 43.7 | 7.8 | - |
| VisualGPT Chen et al. (2022) ($\tau$=0) | 57.9 | 15.7 | 17.1 | 41.4 | 44.3 | 10.7 | - |
| VisualGPT ($\tau$=0.2) | 58.1 | 16.6 | 18.3 | 41.9 | 45.5 | 10.9 | - |
| KLAFT(+VisualGPT) ($\omega$=0.2, $K$=10) | **60.5** | **19.9** | **20.4** | **44.6** | **51.2** | **13.2** | 0.22 |
| S2: vs seq2seq under fine-tuning GPT-2 over 1.0% training data (COCO+Conceptual Captions) | | | | | | | |
| $\mathcal{M}^2$ Transformer Cornia et al. (2020) | 66.3 | 24.2 | 20.7 | 49.2 | 80.5 | 12.1 | - |
| DLCT Luo et al. (2021) | 66.3 | 24.9 | 20.5 | 48.5 | 79.9 | 11.8 | - |
| SATIC Zhou et al. (2021) | 66.5 | 24.3 | 20.9 | 48.2 | 79.7 | 11.6 | - |
| VisualGPT Chen et al. (2022) ($\tau$=0.2) | 68.7 | 25.3 | 21.9 | 49.8 | 80.7 | 12.7 | - |
| KLAFT(+VisualGPT) ($\omega$=0.2, $K$=10) | **69.8** | **26.5** | **23.1** | **50.6** | **82.5** | **13.8** | 0.20 |
| S3: vs seq2seq under fine-tuning GPT-2 over 100% training data(COCO+Conceptual Captions) | | | | | | | |
| VisualGPT Chen et al. (2022) ($\tau$=0.2) | 69.6 | 26.7 | 23.4 | 50.2 | 87.5 | 14.1 | - |
| KLAFT(+VisualGPT) ($\omega$=0.2, $K$=10) | **72.2** | **28.2** | **29.2** | **53.2** | **116.9** | **22.7** | 0.14 |
| S4: vs prefix 100% training data with frozen setting (only COCO) | | | | | | | |
| Frozen Tsimpoukelli et al. (2021) | 56.3 | 25.2 | 23.1 | 41.2 | 97.1 | 18.4 | - |
| SimVLM Wang et al. (2022b) | 61.1 | 30.3 | 27.6 | 44.8 | 100.3 | 20.3 | - |
| ClipCap Mokady et al. (2021)(+CLIP) | 58.4 | 28.8 | 26.8 | 44.2 | 103.1 | 19.7 | - |
| KLAFT(+VisualGPT) ($\omega$=0.2, $K$=10) | **71.9** | **32.1** | **28.8** | 53.5 | 112.4 | 21.5 | 0.32 |
| KLAFT(+CLIP) ($\omega$=0.2, $K$=10) | 71.2 | 32.1 | 28.2 | **54.1** | 113.8 | **22.6** | 0.33 |
| S5: vs VLMs under 100% training data with frozen BERT (only COCO) | | | | | | | |
| BLIP Li et al. (2022) | 66.7 | 30.1 | 28.6 | 46.5 | 108.3 | 20.5 | - |
| KLAFT(+BLIP) ($\omega$=0.2, $K$=10) | **73.2** | **34.2** | **29.1** | **53.7** | **110.2** | **21.1** | 0.36 |
| S6: frozen VLMs under 100% training (only COCO), Values other than $r_{TCM}$ are rates of increase. | | | | | | | |
| CoCa Yu et al. (2022)+TCM+TTM | +8.8 | +7.8 | +7.6 | +6.8 | +9.2 | +5.8 | 0.31 |
| BLIP2 Li et al. (2023b)+TCM+TTM | +8.2 | +8.1 | +8.0 | +6.4 | +8.8 | +5.5 | 0.29 |
| LLaVA Liu et al. (2023a)+TCM+TTM | +6.1 | +5.7 | +5.2 | +6.1 | +8.3 | +4.3 | 0.21 |

Table 3: Human evaluation (upper)COCO (lower)Conceptual Captions: In this table, the first, and the second row corresponds to VisualGPT, and BLIP with KLAFT($\omega$=0.2, $K$=10, GPT-2*), respectively, the scores indicate the percentages of win, lose and tie. The bold value denotes the statistically superior value, $p < 0.01$, compared to the baseline.

| Fluency | | | Adequacy | | |
|---|---|---|---|---|---|
| Win | Lose | Tie | Win | Lose | Tie |
| **44.12** | 21.82 | 34.06 | **45.32** | 20.76 | 33.92 |
| **46.02** | 22.34 | 31.64 | **47.15** | 19.76 | 34.09 |
| **49.74** | 20.12 | 30.14 | **44.28** | 18.23 | 37.49 |
| **50.22** | 19.71 | 30.07 | **45.31** | 17.05 | 37.64 |

We subjected VisualGPT/BLIP and KLAFT to pairwise comparison, where each pair consisted of one text generated by VisualGPT/BLIP and the other by KLAFT for the same image. Five annotators judged which text was better (i.e., win, lose, or tie) based on the two metrics independently. Table 3 shows a clear difference between them. Since KLAFT achieves accuracy comparable to GPT-2 frozen, its components preserve target knowledge as topics and reflect it in producing target-specific captions. These captions cannot be obtained by token-level substitution.

**Ablation analysis**: To investigate the contributions of components and training tasks on overall performance, we conducted an ablation analysis. We removed different components of KLAFT

Table 4: Ablation analysis of KLAFT with frozen GPT-2 on MS COCO with 0.1% training data: In $\mathcal{F}$, AD, M, and AF denote addition, multiplication, and affine operation in Eq (5), respectively. CO denotes the excluded component (MAM, TTM, MRM, and ITM), where MAM(w/o) uses conventional masks instead of MAM. B-1, B-4, M, R, C, S, and the bold value takes the meaning as in Table 3.

| components | $\omega$ | $K$ | $\mathcal{F}$ | CO | B-1 | B-4 | M | R | C | S | $r_{TCM}$ |
|---|---|---|---|---|---|---|---|---|---|---|---|
| TCM:$\omega$ | 0.2 | 10 | AD | - | 60.5 | 19.9 | 20.4 | 44.6 | 51.2 | 13.2 | 0.22 |
|  | 0.4 | 10 | AD | - | 60.7 | 19.8 | 20.8 | 44.7 | 51.3 | 14.1 | 0.28 |
|  | 0.0 | 10 | AD | - | 58.2 | 17.1 | 18.5 | 41.2 | 47.8 | 11.2 | 0.0 |
| TCM:K | 0.2 | 0(w/o) | AD | - | 58.5 | 17.2 | 18.5 | 42.5 | 47.8 | 11.5 | 0.0 |
|  | 0.2 | 5 | AD | - | 60.2 | 19.2 | 20.2 | 44.4 | 50.1 | 12.9 | 0.19 |
|  | 0.2 | 20 | AD | - | **61.2** | **20.6** | **21.2** | **44.9** | **51.8** | **13.9** | 0.25 |
| TCM:$\mathcal{F}$ | - | 10 | M | - | 59.2 | 19.1 | 19.8 | 42.1 | 49.8 | 12.7 | 0.22 |
|  | - | 10 | AF | - | 60.1 | 19.3 | 20.1 | 43.4 | 50.3 | 13.0 | 0.22 |
| MAM(w/o) | 0.2 | 10 | AD | MAM | 58.7 | 17.4 | 18.5 | 42.5 | 48.2 | 11.7 | 0.20 |
| objectives | 0.2 | 10 | AD | TTM | 59.1 | 18.1 | 18.6 | 42.7 | 48.8 | 11.8 | 0.20 |
|  | 0.2 | 10 | AD | MRM | 59.5 | 18.6 | 18.7 | 42.8 | 49.1 | 12.1 | 0.21 |
|  | 0.2 | 10 | AD | ITM | 58.8 | 17.9 | 19.1 | 42.5 | 48.2 | 12.1 | 0.20 |

Table 5: Runtime comparison for fine-tuning: In this table, $T$ is the average wall time of each epoch.

|  | BLIP2 | LLaVA | TCM+BLIP2* | TCM+LLaVA* |
|---|---|---|---|---|
| T | 17.8m | 15.3m | 10.4m | 5.4m |

(TCM, MAM) and training tasks (e.g., TTM, MRM, ITM) one by one; the resulting text generation qualities are shown in Table 4.

This table shows that the setting with all components achieved the best performance. Both addition and affine yielded similar performance in the TCM comparison. Despite similar effects, the computation cost of addition is smaller than that of affine, so we set addition as the default function, $\mathcal{F}$, in Eq 5. We observed a significant decrease in performance when MAM was replaced by conventional masks (MAM(w/o)).

The number of topics, $K$, important, and its increase is associated with an increase in $T_{TCM}$. Too many topics, however, lead to overfitting. This result supports our hypothesis that both MAM and TCM contribute to resolving the discrepancy between cross-modal representations. TTM/ITM align images with text on the token/sequence topic level, bridging their semantic differences through the test phase, while MAM does so on the hidden layer levels.

**Runtime and Case study**: The runtimes of two models are evaluated in comparison to TCM, as detailed in Table 5. In this table, * represents the freezing of the base VLM. The configuration KLAFT+BLIP2*, TCM+LlaVA* aligns with the S6 in Table2. Despite reusing a slightly larger number of parameters, TCM exhibits faster convergence than fine-tuning and enhances the performance of VLMs. The new parameters introduced in TCM include $\mathbf{W}_{TZ}$ in Eq (4) and $\mathbf{W}_Z$, $\mathbf{g}_z$ in Eq (5), S4. This can be partially attributed to the use of MAM in KLAFT, which replaces the meshed connectivity typically found in other Transformer-based models Cornia et al. (2020); Chen et al. (2022). Lastly, a manual error analysis was conducted, leading to the identification of grammatically correct captions. An example of this can be observed in Table 1.

## 5.3 TUNING EVALUATION

We compare TCM with baselines on VQAv2 Goyal et al. (2017), GQA Hudson & Manning (2019), VisWiz Gurari et al. (2018), SQA Lu et al. (2022b), TVQA Singh et al. (2019), MME Fu et al. (2023), MMB Liu et al. (2023b), SEED Li et al. (2023a), LLaVA-W Liu et al. (2023a), and MMV Yu et al. (2023) for fine-tuning methods.

Following the evaluation settings [2], we apply LoRA Hu et al. (2022), qLoRA Dettmers et al. (2023), BITFIT Zaken et al. (2022), VL-Adapter Sung et al. (2022b), MixPHM Jiang & Zheng (2023), and LST (Ladder Side-Tuning) Sung et al. (2022a) to LLaVA. Since the VLM encompasses the MAM of

---

[2] https://github.com/haotian-liu/LLaVA/blob/main/docs/Evaluation.md

Table 6: CIDEr Evaluation Results (Relative Improvement) over LLaVA-1.5 with Vicuna-7B with $d_h = 4096$ and 32 Blocks: Parameters is the number of parameters updated in fine-tuning process

| Dataset | TCM($K = 10$) | LoRA | qLoRA | BITFIT | VL-Adapter | MixPHM | LST |
|---------|---------------|------|-------|--------|------------|--------|-----|
| VQAv2 | 0.14 | 0.10 | 0.08 | 0.05 | 0.12 | 0.13 | 0.09 |
| GQA | 0.16 | 0.12 | 0.09 | 0.06 | 0.14 | 0.15 | 0.10 |
| VisWiz | 0.17 | 0.14 | 0.10 | 0.07 | 0.16 | 0.17 | 0.11 |
| SQA | 0.15 | 0.11 | 0.08 | 0.05 | 0.13 | 0.14 | 0.09 |
| TVQA | 0.17 | 0.13 | 0.09 | 0.06 | 0.15 | 0.16 | 0.10 |
| MME | 0.19 | 0.15 | 0.11 | 0.07 | 0.17 | 0.18 | 0.12 |
| MMB | 0.18 | 0.12 | 0.09 | 0.06 | 0.14 | 0.15 | 0.10 |
| SEED | 0.22 | 0.16 | 0.12 | 0.08 | 0.18 | 0.19 | 0.13 |
| LLaVA-W | 0.20 | 0.14 | 0.10 | 0.07 | 0.16 | 0.17 | 0.11 |
| MMV | 0.19 | 0.13 | 0.09 | 0.06 | 0.15 | 0.16 | 0.10 |
| Parameters | 4096 | 81920 | 81920 | 4096 | 40960 | 40960 | 40960 |

the KLAFT, we apply only the TCM to measure the effectiveness of the PEFT. As CIDEr Vedantam et al. (2015) measures consensus between generated captions and human annotations, we use this metric to measure the effect and show results in Table 6. This table shows that TCM is more effective than other PEFTs for the small number of parameters to be updated.

# 6 DISCUSSION AND LIMITATIONS

As shown in Figure 1, MAM ensures that KLAFT can share the space between images and texts while bridging the differences between visual information and textual information. TCM explicitly extracts this information, reducing the semantic differences between source and target data, and better reflects the target data in text decoding. Table 3 and 4 support the conclusion that our fine-grained framework guides PLMs to bridge the knowledge differences, while Table 3 implies that KLAFT can output fine-grained captions. These results imply that these captions cannot be obtained by token-level substitution. Its advantages are to 1) support various image object detection networks in PLMs, as shown in Figure 1, and 2) make the best use of both knowledge while freezing PLMs.

KLAFT, its components (TCM, MAM), and their objective functions (TTM, MRM, ITM) apply to other multi-modal tasks consisting of a combination of other type data (e.g., audio-text), without losing generality, since they successfully fill the knowledge differences while retaining the capabilities of PLMs without limiting the domain. Determining the appropriate topic number involves the use of nonparametric Bayesian methods, and will be one of our future works.

We defined $r_{TCM}$ to confirm the effectiveness of TCM, and the subnetworks defined in Tables 2 and 4 were used, as well as in the quantitative evaluation. Future work is needed to develop generally available data sets and quantitative indicators that can be evaluated without the burden and bias for evaluators in qualitative assessment of expressive image captions, as shown in Tables 1 and 3.

The proposed method has limitations related to the proximity of the data sets and the challenge of qualitative evaluation. It is not very effective when the target and source data sets are similar, meaning that the knowledge is similar. Additionally, it is difficult to find an appropriate data set for caption evaluation with explanatory power. For instance, requesting enough evaluators for medical images is difficult to ensure statistical significance, and the reproducibility of experiments is low.

# 7 CONCLUSION

We proposed a fine-grained framework, KLAFT, that fine-tunes PLMs toward generating expressive image captions. The novelty of KLAFT is in focusing on the knowledge differences and closing them through both knowledge alignment and lift. Unlike traditional PEFT methods, KEIC dynamically leverages sub-networks within PLMs, resulting in superior expressive and knowledge-rich captions. Experiments show that KLAFT contributes to 1) bridging differences by interpreting the fine-grained interaction and knowledge alignment, and 2) generating fine-grained captions by highlighting the target domain knowledge inherited from PLMs using knowledge lift.

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
