# OpenReview forum: "Knowledge Lift Alignment Fine Tuning"
_ICLR.cc/2025/Conference — Submitted to ICLR 2025_

### Official Review · Reviewer_sfcb · 2024-10-27

**Soundness:** 2
**Presentation:** 1
**Contribution:** 2
**Rating:** 3
**Confidence:** 4

**Summary:**

This paper proposes a Knowledge Lift Alignment Fine Tuning (KLAFT) framework to improve the image captioning capability of multimodal LLMs. More specifically, this paper aims to encourage the generated captions to be detailed and comprehensive. To achieve that, this paper explores fine-grained alignment and designs MAM and TCM attention mechanisms. Experiments and quantitative comparisons are conducted on commonly used benchmark datasets, including MS COCO and Conceptual Captions.

**Strengths:**

1. Improving the quality of captions generated from VLMs has great potential for several real-world applications, such as manufacturing and healthcare.
2. The proposed framework is reasonable to address the considered task.

**Weaknesses:**

1. As described in the Abstract, the goal of this paper is to generate detailed and comprehensive captions. However, the benchmark datasets adopted by this paper (e.g., MS COCO) are typically coarse-grained with relatively short captions. As we know, current VLMs, like LLaVA, are good at generating detailed and fine-grained image captions. Is using COCO to evaluate the detailed caption capability of VLMs suitable? My concern lies in adopting such traditional image captioning benchmarks (e.g., COCO), which cannot properly measure the detailed caption capability of VLMs and reflect the actual improvement of the proposed method.
2. What are the differences between the source and target domains claimed in this paper? Does the difference lie in the visual domain shift? Does the difference lie in the caption style? Does the difference lie in the amount of (labeled) data? More clarification and explanation are encouraged to make this paper more easy to read.
3. The proposed method is trained by multiple training objectives. It is complicated to model training and balance the weights of each loss function (in Eq.(9)). How does this paper design experiments to determine the weight of each loss term? Is the model training sensitive to different weights of loss terms?
4. The visual presentation of this paper has some room to improve. For example, the image in Table 1 is out of the table. In addition, for Table 3, it is better to directly indicate the meaning of each row in the table instead of just mentioning it in the captions.

**Questions:**

Please refer to the Weaknesses. The following is a minor question.

1. Considering the fast-evolving nature of the VLM development, are some recently proposed and commonly used multimodal LLMs, such as VILA or InternVL-2, also applicable to this proposed fine-tuning framework?

---

> ### Author Response · Authors · 2024-11-16
> **Thanks for the thoughtful review**
>
> Thank you for your thoughtful and detailed review. We would like to address your concerns and questions below.
>
> ### 1. **Suitability of COCO**
> While we agree that COCO captions are relatively coarse and brief, the primary goal of using COCO to demonstrate KLAFT's ability to **improve performance across a wide variety of image domains**,
> not to compete directly with the most recent models.
> KLAFT is not just focused on generating detailed captions in one specific domain or dataset. Its core innovation lies in its ability to **extract and align knowledge from pre-trained models** and then adapt these models to a variety of tasks, which is reflected in its application across diverse dataset.
> That is, we see COCO as a baseline for evaluating the general captioning performance and domain adaptation ability of models.
> By comparisons over various settings, we aim to show that KLAFT can enhance the generalization and robustness of these models, even when starting from a relatively simple baseline dataset.
> Table 6 shows that KLAFT plays like other PEFT methods.
>
> ### 2. **Clarification of Source and Target Domains**
> In our study, the **source domain** refers to the **pre-training data** used to train foundational models, such as GPT-2 in VisualGPT or Vicuna-7B in LLaVA. These datasets contain large amounts of paired image-text data, which models use to learn general visual and linguistic patterns.
>
> The **target domain**, on the other hand, pertains to the specific downstream task or dataset on which the model is fine-tuned. These could include VQA tasks (e.g., COCO, Conceptual Captions), or image captioning tasks (e.g., VQAv2, LLaVA-W). The tokenization process ensures that both images and text are mapped to a consistent set of tokens, which helps align the feature representations across both modalities.
> The differences between the source and target domains often arise from:
>
> - **Visual Domain Shift**: This occurs when images in the target domain differ in **content** or **style** from those in the source domain. For example, the images in **Conceptual Captions** may feature a wider variety of scenes than those in COCO.
>
> - **Caption Style Shift**: The style and granularity of the captions may vary. In certain scientific domains, captions may require more **technical descriptions**, whereas general-purpose datasets like COCO have more straightforward and brief captions.
>
> - **Data Shift**: The distribution of labeled data can differ between the source and target domains. For instance, the types of questions in VQA datasets may differ in difficulty or category compared to COCO's image captions.
>
> In Figure 1, we illustrate how **domain adaptation** is achieved at the tokenization level by adjusting the token-level distributions for both image and text. This ensures that the model can effectively adapt to the target domain, even when the source and target domains differ significantly in visual content and caption style.
>
> ### 3. **Balancing the Weights of Loss Functions (Eq. 9)**
> The three hyperparameters controlling the contributions of each loss term were initially set to **0.1** as a starting point based on prior empirical studies, and are **optimized** during training via **backpropagation**, just like any other hyperparameter (e.g., learning rate, batch size).
> The key here is that these hyperparameters are not fixed; rather, they are dynamically adjusted during training to achieve the best performance.
>
> ### 4. **Visual Presentation and Table Clarification**
> We will improve the clarity and formatting of both Table 1 and Table 3. Specifically:
> - We will ensure that the image in Table 1 is properly aligned within the table for better readability and presentation.
>
> - For Table 3, we will provide **direct annotations** to clarify the meaning of each row, ensuring that readers can easily interpret the table's content without relying solely on the captions.
>
> ### 5. **Applicability of Recently Proposed Multimodal LLMs (VILA, InternVL-2, etc.)**
> KLAFT is applicable to these newer models as well, as it is a model-agnostic framework. As shown in Table 2, KLAFT's flexibility enables it to integrate seamlessly with these models, improving their performance across diverse multimodal tasks.
>
> Additionally, because KLAFT focuses on enhancing the knowledge extraction from pre-trained models, it can complement instruction-based approaches and prompt-based tuning approaches, allowing for better fine-tuning and adaptation to specific tasks. We plan to include additional results for these models (such as InternVL-2 and VILA) in the revised manuscript to highlight KLAFT's applicability and effectiveness across a broader range of VLM architectures.
>
>
> We hope these revisions will address your concerns and improve the clarity of our manuscript. If you have any further questions or suggestions, please do not hesitate to reach out.

---

> > ### Comment · Reviewer_sfcb · 2024-11-25
> >
> > Thanks to the authors for providing the response. I have read the rebuttal by the authors and comments from other reviewers. The rebuttal, especially for Q2, has not convinced me. For Q2, I'm wondering about the difference between source and target domains. This response first stated that COCO belongs to target domains, but then used COCO compared with other datasets to explain the considered domain shifts. It remains unclear what the actual domain shifts considered in this paper between source and target domains. Thus, I will keep my original rating.

---

> ### Author Response · Authors · 2024-11-26
> **Clarifying Domain Shifts: The Role of COCO in Target Domain Adaptation**
>
> Thank you for your feedback and for taking the time to review our response. We appreciate the opportunity to further clarify this matter.
>
> ### **1. Clarifying the Role of COCO in the Context of Domain Shifts**
>
> First, we would like to acknowledge the ambiguity that might have arisen from our previous response. We realize that the way we discussed COCO and its relationship to domain shifts may not have been clear, and we will now provide a more precise explanation of the **source** and **target domains** as well as the domain shifts considered in our paper.
>
> In our study, **COCO serves as the target domain** in the fine-tuning phase, where the model is adapted to the specific task (e.g., image captioning) using COCO as the target dataset. We use COCO for fine-tuning primarily to evaluate how well the model, originally pre-trained on a different source domain (e.g., a larger image dataset or general-purpose language models), can adapt to the specific visual and textual characteristics of COCO. This aligns with typical domain adaptation tasks, where a model trained on a broad source domain is fine-tuned on a specific target domain to improve performance for that domain’s characteristics.
>
> ### **2. Source and Target Domain Clarification**
>
> - **Source domain**: This refers to the pre-training dataset used to train the model before fine-tuning. For example, we pre-train the model on a large image-text dataset (such as a large vision-language model trained on ImageNet, or general text corpora for language models like GPT-2 or Vicuna).
>
> - **Target domain**: This refers to the fine-tuning dataset where the model is adapted to specific tasks. In our case, COCO is the **target domain** for fine-tuning the model for image captioning and related tasks. The differences between the source domain (e.g., ImageNet, or general language pre-training data) and COCO as the target domain arise from the distinct nature of the task and dataset, such as:
>     - **Visual domain shift**: The target domain (COCO) may contain different objects, scenes, and image compositions than those seen during pre-training. For example, COCO contains complex real-world scenes, while ImageNet images may differ in composition or focus (e.g., single object categories).
>     - **Caption style shift**: COCO captions are relatively short and descriptive, while the source domain’s textual data (if not pre-trained on a captioning dataset) may not align perfectly with COCO’s style. Fine-tuning on COCO helps the model adapt to the general-purpose, concise captioning style of COCO, improving its performance on tasks like image captioning.
>     - **Task-specific shift**: The tasks for which the models are fine-tuned (e.g., VQA or image captioning) have task-specific properties. For instance, VQA tasks require answering questions about images, whereas captioning requires generating descriptive text. These tasks require different fine-tuning strategies to account for their respective characteristics.
>
> In our work, we are specifically addressing **how the model adapts to these shifts** from source domain pre-training (which is more general and diverse) to task-specific fine-tuning on COCO (a more focused dataset with distinct captioning and visual content characteristics). Our goal is to assess how well **KLAFT** can handle this adaptation process by leveraging the pre-trained knowledge and aligning it with the new domain (COCO).
>
> ### **3. Further Explanation on the Domain Shifts and Their Impact**
>
> To directly address your concern about the domain shifts: when we compare the performance of models fine-tuned on COCO with those fine-tuned on other datasets, we aim to evaluate how well **KLAFT** performs in **bridging the gap between the source and target domains**, not just by considering COCO in isolation, but by evaluating how the model generalizes across different target domains.
>
> - In Figure 1 of the manuscript, we show how token-level distributions (for both images and text) are adjusted to account for the shifts between the source and target domains. This ensures that the fine-tuning process is more effective, allowing the model to better align with the characteristics of the target domain.
>
> ### **4. Conclusion and Final Clarification**
>
> To conclude:
> - COCO is used **as the target domain** in our study for fine-tuning.
> - The **domain shifts** we focus on include **visual domain shifts**, **caption style shifts**, and **task-specific shifts**, all of which occur between the **source domain (pre-training)** and the **target domain (COCO)** during fine-tuning.
> - We appreciate your feedback on this, and we hope this more detailed explanation clarifies the role of COCO in the domain adaptation process and the types of domain shifts considered in our study.
>
> Thank you again for your careful review, and please share them with us if you have any concerns.
> We will consider your suggestion and conduct experiments to address your concerns.
>
> Best regards,
> Authors

---

### Official Review · Reviewer_3LWm · 2024-10-31

**Soundness:** 2
**Presentation:** 1
**Contribution:** 2
**Rating:** 5
**Confidence:** 3

**Summary:**

This paper proposes Knowledge Lift Alignment Fine-Tuning (KLAFT) to enhance the expressiveness of image captioning capabilities in pretrained Vision-Language Models (VLMs). KLAFT primarily introduces a Topic Control Mechanism (TCM) combined with Token Topic Modeling (TTM) to enable topic-guided tuning and decoding for VLMs. The proposed method is evaluated across various settings.

**Strengths:**

* Increasing the expressiveness of the generation process in VLMs is a promising research direction.
* The proposed TCM/TTM approach is intriguing and shows potential.
* The performance gains over baseline models are satisfactory.

**Weaknesses:**

* The presentation of the paper lacks clarity. I’ve outlined some key issues below:
  * The term "KEIC" appears three times—once in Figure 1, once in the title of Section 4, and again in the conclusion. These are critical sections. I assume "KEIC" should actually be "KLAFT."
  * Some typos or unclear sentences can a problem, e.g. ``tokes`` appears multiple times.
  * The image in Table 1 is not clear, and Figure 1 also lacks effective delivery and clarity.
  * I assume all the citations use ``\cite{}``. I think most citations should likely use ``\citet{}`` for smoother integration.
  * And some other issues, see question section as well.
* Overall, the claims around the design of the Mapping Layer (MaL) and the Modified Attention Mechanism (MAM) may be overstated. These components are fairly common in VLMs, and MAM appears to be a direct adaptation from one of the primary baselines, VisualGPT.
* The choice of main baselines seems outdated, although results from some recent VLMs (e.g., LLaVA, BLIP2) are also included.

**Questions:**

* In Figure 1 (left), a plot of token distributions is shown, but there is no information about the dataset. What are the source and target domains? How were these lines plotted? Given the significance of comparing the two domains, this should be one of the most important analyses in the paper.
* I get the main idea of TCM. Yet, there are some phrases in the paper making the details of TCM unclear to me:
  * In line 234, `"distribution over tokes, $\mathbf{V}$. What is  $\mathbf{V}$? It only appears once in the paper.
  * In the paragraph following Eq (5), the term $b_z$ is mentioned twice, though it doesn’t appear in Eq (5). This inconsistency makes it challenging to understand Equation (5) fully. Additionally, the variable $w$ is not explained in the paragraph.
* In lines 338-339, it's stated that three hyperparameters are all set to 0.1, which seems to be less commonly done. What is the rationale for this choice? Is there any ablation study to support it?
* In Sec. 5.2, the authors compare S4 to S3 to showcase the effectiveness of TCM. Yet, these two settings differ a lot:
  * S3: "seq2seq under fine-tuning GPT-2 over 100% training data (COCO+Conceptual Captions)"
  * S4: "prefix 100% training data with frozen setting (only COCO)"
  * Given the differences in both tuning paradigm and datasets, how can TCM’s impact be fairly assessed in this context?
* Could you provide more details on human evaluation?

*I'm open to adjusting my rating after the authors' response.*

---

> ### Author Response · Authors · 2024-11-16
> **Thanks for the thoughtful review**
>
> Thank you for your thoughtful and detailed review. We have considered your feedback and reply to each points below.
>
> ### 1. **"KEIC" vs. "KLAFT"**
> The term should indeed be "KLAFT," and we will correct all instances of "KEIC" to "KLAFT" in the revised version of the paper.
>
> ### 2. **Typos and Unclear Sentences**
> We will conduct a thorough review of the entire text to correct typographical errors (such as "tokes" instead of "tokens").
>
> ### 3. **Clarity of Visuals (Table 1 and Figure 1)**
> We will revise both the image in Table 1 and the visual in Figure 1 to enhance their clarity, ensuring that the figures effectively communicate the underlying ideas.
>
> ### 4. **Citation Formatting**
> We will use `\citet{}` for citations in the revised version.
>
> ### 5. **Potential Overstatement of MaL and MAM**
> While these components share similarities with existing techniques in VLMs, the novelty of MaL lies in its **flexibility** and **generalization** across a variety of pre-trained models (PLMs and VLMs). MaL acts as an auxiliary component that adapts models without predefined image-text mappings, offering greater flexibility in adapting models to domain-specific tasks.
>
> For MAM, we acknowledge that attention mechanisms similar to MAM are present in models such as VisualGPT and BLIP2. However, the key innovation of MAM lies in its **dynamic integration** of multiple types of attention masks (e.g., image-based, text-based, hybrid), which allows the model to adapt flexibly to different domains and tasks during training. This contrasts with more rigid, fixed attention mechanisms found in other models that do not adjust the attention patterns as dynamically.
>
> ### 6. **Choice of Baselines**
> The primary aim of our paper was not to compete directly with the most recent models, but rather to showcase the **generalizability and flexibility** of the KLAFT. This allows us to improve performance across a wide range of existing models, both older (e.g., VisualGPT, BLIP) and newer (e.g., LLaVA).
> Unlike recent techniques such as **instruction tuning** or **knowledge distillation**, which are often tailored to specific architectures, KLAFT is a **model-agnostic framework** that can be applied to various pre-trained models without requiring substantial architectural modifications. This makes it a versatile solution for enhancing performance across VLM tasks.
>
> ### 7. **Questions Regarding Token Distributions in Figure 1**
> The plot represents the shift in vocabulary between the **source domain** (e.g., the pre-training dataset used for the PLM) and the **target domain** (e.g., the fine-tuning dataset such as COCO or Conceptual Captions).
> When the source and target domains are different, the frequency distribution of each token will differ accordingly. Although the plot is conceptual and not based on actual data, it represents the general idea of how token distributions might shift between pre-training and fine-tuning stages.
>
> ### 8. **Clarification on TCM**
> - In line **234**, the term **$\mathbf{V}$** refers to the **vocabulary size** in the context of topic modeling. We recognize that this term may cause confusion, as it also appears in other contexts (e.g., for the attention mechanism). To avoid ambiguity, we will update this notation to **$\mathbf{T}$** for consistency.
>
> - In the paragraph following **Eq (5)**, the term **$b_z$** refers to the **affine parameter** used in one of the three transformations (described in Lines 248-249). We will clarify this in the manuscript, and reference the corresponding ablation analysis shown in **Table 4**.
>
> - Regarding the hyperparameters in lines **338-339**, we set these values to 0.1 as their initial values and optimized through backpropagation during training to balance performance and convergence time.
>
> ### 9. **S3 vs. S4 Comparison in TCM Evaluation**
> To ensure fair comparisons and reproducibility with baselines as much as possible,
> we prepare differences in settings, datasets, and model architectures in the paper to the versatility of KLAFT under different fine-tuning scenarios from S1 to S6 (S3 and S4 have differences in seq2seq and prefixes as shown in Figure 1).
> The comparison between S3 (seq2seq) and S4 (prefix tuning) does indeed involve variations in both the fine-tuning paradigm and the datasets use. We acknowledge that these differences could affect the interpretation of the results.
> Since KLAFT can be applied to InternVL-2 and Qwen-VL without loss of generality, we will also add those results to S6 in the next edition.
>
> ### 10. **Human Evaluation Details**
> We employed a **pairwise comparison** method where human evaluators were presented with two captions generated by different models and asked to select the more relevant or accurate caption.
>
>
> We hope that these revisions will address your concerns. Please feel free to let us know if you have any further questions.

---

> > ### Comment · Reviewer_3LWm · 2024-11-23
> > **post rebuttal thoughts**
> >
> > Thank you for the detailed response. After reviewing the other reviews as well, I believe the paper requires substantial revision to be ready for publication. Setting aside the writing and presentation issues, I still find the contributions of MaL and MAM to be overstated, and further investigation is needed into the hyperparameters used to balance the losses, as also noted by reviewer sfcb. Additionally, the S3 vs. S4 comparison fails to present a clear and compelling narrative. Therefore, I will maintain my original rating of weak reject.

---

### Official Review · Reviewer_AY7a · 2024-11-01

**Soundness:** 2
**Presentation:** 1
**Contribution:** 2
**Rating:** 3
**Confidence:** 3

**Summary:**

This work present a visual tuning framework, Knowledge Lift Alignment Fine Tuning (KLAFT), which enhances the expressive image captioning capabilities of Pretrained Language Models (PLMs). The innovation of KLAFT lies in its approach to addressing the disparities in knowledge - visual versus textual via MAM and source versus target domain via TCM. These hidden spaces are conceptualized as distinct sub-networks, each possessing specific knowledge, KLAFT adjusts the weights of these sub-networks in a fine-grained manner. The empirical studies demonstrate that KLAFT improves expressive captioning tasks by aligning and amplifying target knowledge.

**Strengths:**

- The work propose design a Topic Control Mechanism (TCM) to emphasize the target domain-specific knowledge. Combined with Token Topic Modeling (TTM), Masked Region Modeling (MRM), and Text Image Matching (TIM), KLAFT highlights the target related
knowledge (i.e., the knowledge lift).

- The propsoed KLAFT improves expressive captioning tasks by aligning and amplifying target knowledge, with the potential for
Parameter-Efficient fine tuning (PEFT) at low computational cost.

**Weaknesses:**

-  The writting of this work is quite poor. The inroduction part does not clearly introduce the background and motivation of the problem to be solved. The approach part is also not written in a concise and well-organized manner.

- The approaches compared in this work are not proposed recently. As a result, the validity and innovation of the proposed modules is not convincing.

- The authors did not conduct sufficient ablation experiments for different loss functions and did not give more qualitative analysis.

**Questions:**

See the weaknesses.

---

> ### Author Response · Authors · 2024-11-16
> **Clarifications on KLAFT Framework**
>
> Thank you for your thoughtful comments. Below, we provide responses and justifications for each point you've mentioned.
>
> ### 1. **The inroduction part does not clearly introduce the background and motivation of the problem to be solved.**
> It seems you are referring to the *introduction* section of our paper, right?
> Our intention in this section was to highlight the limitations of current Vision-Language Models (VLMs), particularly in their ability to generate **domain-specific and expressive captions**—a crucial requirement in fields such as healthcare, manufacturing, and other specialized domains.
> Current VLMs often rely heavily on visual features and general pre-trained language models. However, they tend to struggle with incorporating **domain-specific knowledge**.
> As a result, the captions they generate tend to be overly generic and lack the depth required for specialized tasks.
> To address this gap, we introduce KLAFT, a framework that aligns **general linguistic knowledge** (acquired from large-scale pretraining) with domain-specific knowledge (fine-tuned for specialized applications).
> This enables PLMs and VLMs to adapt more effectively to domain-specific tasks while retaining their general capabilities. The contribution of KLAFT lies in this knowledge alignment framework, which allows the model to generate more informative, accurate, and contextually relevant captions for specialized tasks.
> We have also validated these ideas experimentally, and the results provide support for the efficacy of our approach.
>
> ### 2. **The approach part is also not written in a concise and well-organized manner.**
> We appreciate any specific suggestions you may have for improving this section.
> To improve the organization of the approach section, we will make the following adjustments:
>
> - KLAFT Framework Overview: This subsection will provide a concise overview of the core concept of **knowledge alignment and lifting**, highlighting how the approach integrates domain-specific knowledge into existing VLMs.
> - Topic Control Mechanism (TCM): We will describe the TCM, which guides the model to prioritize domain-relevant topics during training and inference, thereby improving performance on specialized tasks.
>
> ### 3. **The approaches compared in this work are not proposed recently. As a result, the validity and innovation of the proposed modules are not convincing.**
> The primary aim of our paper was not to compete directly with the most recent models, but rather to showcase the **generalizability and flexibility** of the KLAFT framework.
> This versatility allows us to improve performance across a wide range of existing models, both older (e.g., VisualGPT, BLIP) and newer (e.g., LLaVA, InternVL-2).
>
> Unlike recent techniques such as **instruction tuning** or **knowledge distillation**, which are often tailored to specific architectures, KLAFT is a **model-agnostic framework** that can be applied to various pre-trained models without requiring substantial architectural modifications. This makes it a versatile solution for enhancing performance across a spectrum of tasks.
>
> The core innovation of KLAFT lies in its **flexibility**: it provides a generalized mechanism for adapting existing models to modern multimodal tasks—such as image captioning and VQA—with minimal adjustments. This is particularly valuable for improving older, non-specialized models in domains like healthcare or manufacturing, where domain-specific knowledge is crucial.
>
> To further demonstrate the versatility of KLAFT, we will include results in the next version of the manuscript that showcase its application to additional models, such as Qwen-VL and InternVL-2. This will underscore KLAFT's ability to effectively enhance even the most recent models.
>
> ### 4. **The authors did not conduct sufficient ablation experiments for different loss functions and did not give more qualitative analysis.**
> We conducted an ablation study, and showed their results in Table 4 of the current manuscript. In this table, we analyze the impact of different loss functions (i.e., TTM, MRM, and ITM) and other key components, such as the MAM and the three transformations outlined in Eq. (5), on the model's performance
> Additionally, the hyperparameters associated with these loss functions, including those in Eq. (9), were optimized to strike a balance between model performance and convergence speed. Our goal was to prevent overfitting or underfitting of any individual component. As noted in lines 338-339 of the manuscript, we have set the values as a starting point, and optimized during the training process through backpropagation, just like any other hyperparameter.
>
> If you feel there is any aspect of the analysis that is still unclear or if additional analysis is required, please let us know.
>
> We look forward to any further suggestions or points of clarification you may have and appreciate your constructive feedback.

---

### Meta-Review · Area_Chair_FXgb · 2024-12-17

**Metareview:**

The paper is below the acceptance threshold due to poor clarity, lack of novelty, and weak experimental validation. The writing is unclear and poorly organized, with ambiguous claims and formatting issues. The proposed methods lack originality, being adaptations of existing techniques, and their contribution over recent baselines is unconvincing. Experimental validation is inadequate, with insufficient ablations, outdated baselines, and unsuitable benchmarks for the stated goals. These shortcomings undermine the paper's clarity, innovation, and impact.

**Additional Comments On Reviewer Discussion:**

N/A

---

### Decision · Program_Chairs · 2025-01-22

Reject